# Is Surface Topography Useful in the Diagnosis of Scoliosis? Validation of the Biometrical Holistic of Human Body (BHOHB)

**DOI:** 10.3390/children10020320

**Published:** 2023-02-07

**Authors:** Angelo G. Aulisa, Diletta Bandinelli, Martina Marsiolo, Francesco Falciglia, Marco Giordano, Renato M. Toniolo

**Affiliations:** 1UOC of Traumatology, Bambino Gesù Children’s Hospital, IRCCS, Piazza S. Onofrio 4, 00165 Rome, Italy; 2Department of Human Sciences, Society and Health, University of Cassino and Southern Lazio, Via S. Angelo in Theodice, Località Folcara, 03043 Cassino, Italy

**Keywords:** scoliosis, surface topography, BHOHB, X-rays

## Abstract

Background: The concern around repeated exposure to X-rays has been motivating doctors involved in scoliosis to seek alternative solutions. Surface topography (ST) analysis is a modern system that has been shown to have good results. The purpose of the study is to validate the new BHOHB hardware for the investigation of scoliosis in adolescents by comparing it to X-ray examinations and to assess the reliability of intraoperator and interoperator tests. Methods: Ninety-five patients were enrolled in our study. All the patients were analyzed via the BHOHB method 2 times by 2 independent physicians (t0) and for a second time 2 or 3 months later (t1). The Pearson correlation coefficient was used to evaluate the relationship between the measurements obtained by BHOHB and the gold standard. The intraclass correlation coefficient (ICC) was used to assess intra- and interoperator reliability. Statistical analysis was performed with the GraphPad Prism 8 software. Results: The correlations between the first and second operators in the measurements and between the BHOHB method and X-ray showed a very good to excellent r for both. A very good correlation was also confirmed for prominence measured by operators and by the BHOHB machine. Intra- and interoperator reliability was found to be very positive for both the first and the second physicians. Conclusions: We can state that ST can be useful for diagnosing and treating scoliosis. The recommendation is to use it primarily to evaluate the evolution of the curve, as in this mode, you can reduce the patient’s exposure to X-rays. The results indicate that BHOHB measures are comparable to radiographs and not influenced by the operator.

## 1. Introduction

Scoliosis is a complex structural deformity of the spine involving three space planes: frontal, sagittal, and axial. Eighty per cent of scoliosis cases have an unknown aetiology; therefore, it is called idiopathic. Scoliosis is defined by the SRS (Scoliosis Research Society) and SOSORT (International Society on Scoliosis Orthopaedic and Rehabilitation Treatment) as a curve of at least 10°, according to Cobb’s method of measurement [1].

According to various studies [1,2,3,4,5,6,7,8,9,10,11,12,13,14,15,16,17,18,19], the prevalence of scoliosis in the population ranges between 0.93% and 12%. The most common age at which scoliosis occurs, about 80% of all cases, is in early adolescence (older than 10 years). 

When the deformity is minor, less than 20°, treatment typically includes observation and surveillance to look for indications of curve progression. For deformity that is more pronounced, the treatments indicated are bracing and surgery for curves greater than 20° and 45°, respectively. Management for these patients also includes surveillance for evidence of change in the deformity [1].

Scoliosis, if not adequately treated, can evolve and lead to various diseases during adulthood, with poor quality of life, body deformity, and disability; it can also cause pain and progressive functional limitation [1]. Therefore, when we speak of spinal deformities we should not exclusively focus on the spine but should also include the patient’s appearance and the assessment of the entire musculoskeletal apparatus including soft tissues, muscles, and bony structures. 

The gold standard for measuring the curve angle in degrees is the Cobb method [20], which has an intraobserver and interobserver variability that can range from 3° to 5° [21]. Additionally, as scoliosis is a 3D deformation [22], vertebral rotation is normally measured via the Perdriolle system. Even though the Cobb method is considered the gold standard, the description of scoliosis using only Cobb grades is restrictive. In fact, the coronal deviation of the spine is not the only factor responsible for somatic back mutations: scoliosis can cause thoracic deformation, asymmetry of triangle size, and scapular profile alteration. In addition, scoliosis influences frontal balance and changes the body’s sagittal plane. 

Moreover, prominence is the most evident clinical sign of scoliosis and represents an important prognostic factor in terms of curve progression. However, in recent years, a correlation between clinical deformity and curve severity has been demonstrated [1].

As a result of these limitations, new methods for assessing these variables have been introduced.

As of today, X-ray is the main diagnostic system for the evaluation of idiopathic scoliosis. However, it should be emphasized that X-rays can have dangerous effects; repeated exposure to ionizing radiation can increase the risk of breast cancer in girls with scoliosis [22,23,24]. Furthermore, at low-dose rates, cancer risks substantially improved over the last two decades. Rühm in a recent study showed an increased risk of leukaemia and solid cancers. A dose–risk relationship is clearly demonstrated for diseases of the circulatory system. With in vivo experiments, very low-dose rates have generally been shown to increase life span through a mechanism thought to be adaptive response; an inverse dose rate response appears to exist for mutation and cataract formation [24]. The importance of reducing radiation is also underlined by the recommendations of the International Commission on Radiological Protection of “Re-enforcing the principle of optimisation of protection, which should be applicable in a similar way to all exposure situations, with restrictions on individual doses and risks, namely dose and risk constraints for planned exposure situations…” [25].

However, X-rays can only assess bony structures, and their ability to measure pelvic position and spinal rotation is limited. To measure the deformities of the trunk, it is necessary to use 3D imaging techniques such as magnetic resonance imaging (MRI). MRI scans of the spine do allow a radiation-free assessment of the spine and trunk; however, they are associated with limited availability, high costs, and, in young patients, the necessity for sedation.

To reduce costs and radiological load while enabling functional measurements of the spine and trunk, alternative imaging techniques have been developed, and for these reasons a new harmless method is proposed in this study.

In recent years, optical techniques have shown greater scientific potential than X-ray, as they are capable of evaluating the morphology of the spinal column while minimizing ionizing radiation. Most popular methods use light beams such as the Integrated Shape Imaging System (ISIS) [26], ISIS2 [27], Fotometric 4D [28], the Orthoscan [29], and Shannon’s surface dynamic topography [30], or photographic beams, which use markers for spine morphology reconstruction with a specific software.

One photographic system is the Biometrical Holistic Of Human Body (BHOHB, Figure 1). This is a diagnostic device that detects postural spine imbalances and paramorphisms such as scoliosis, kyphosis, and lordosis and calculates Cobb angle thanks to its distinctive software, SpinalMeter.

The BHOHB is classified as a medical device (I m class) for postural assessment and for evaluation of both the fixed and mobile spine. BHOHB technologies are certified by the Italian Ministry of Health and are patented at European (EP 3225155 A1; application number: 08425006.7) and worldwide levels (PCT/EP2019/082111). This system, with a simple camera, can achieve postural detection and 3D reconstruction of the spine. Simple digital photos are used to obtain a total body postural analysis. The measurements are ensured by a simple software calibration performed before patients’ screening, which is based on the determination of four points on the standard calibration platform. In addition, the software is able to measure and express the spine curve in Cobb degrees.

The primary aim of this study was to validate the new BHOHB technology in adolescents with scoliosis compared to the gold standard X-ray method.

The secondary aim was to evaluate intraoperator and interoperator reliability in the application of BHOHB technology and the correlation of prominence measurements obtained with BHOHB technology and the humpmeter (both in degrees and millimetres).

## 2. Materials and Methods

### 2.1. Study Design

The study was set up as a monocentric cross-section.

### 2.2. Setting and Population

The study was performed at Bambino Gesù Children’s Hospital in Rome, Italy.

Enrolled patient legal curators (parents) were informed with written consent that they had to sign.

From April 2021 to January 2022, 113 consecutive patients affected by idiopathic scoliosis who had undergone an X-ray of the spine in stand-up position at least 2 months before their first examination were included in our study.

#### 2.2.1. Inclusion Criteria

The patients in the study responded to the following criteria:-Age between 10 and 16 years old;-Orthostatic radiography of the spine in 2 standard projections, for diagnostic purposes, at least 60 days before the first exam;-Idiopathic scoliosis with a Cobb angle >10°;-Positive Adams test;-No prior conservative brace treatment.

#### 2.2.2. Exclusion Criteria

Patients were excluded from the study according to the following criteria:-Scoliosis secondary to other illnesses (neuromuscular, congenital, neoplastic, osteochondrodystrophic);-Spine X-ray taken more than 60 days after study start;-Previous spinal surgery;-Adams test negative with curves <10°;-Patients with a BMI (Body Mass Index) higher than 25.0 kg/m^2^.

### 2.3. Protocol Examination

All patients were examined clinically and with BHOHB by 2 independent operators twice: at the beginning of the study (t0) and once more 2 to 3 months later (t1). The exam took a few minutes and was absolutely noninvasive and safe for the patients.

Prominence was measured with the scoliometer (degrees) and the humpmeter (millimetres) in the Adams test position.

In this test, the patient bent forward, starting at the waist, until the observed prominence was parallel with the floor, with the feet together, arms hanging, and the knees in extension. The palms were held together and placed between the knees.

The operator moved the scoliometer up and down the prominence until the largest ATR measurement was achieved and likewise was evaluated with the humpmeter.

First, the analysis involved positioning the patient on a special platform (Figure 2) after placing 11 markers in predefined spots on the patient’s back (Figure 3).

The set points can be palpated on the patient’s body as follows:C7 vertebral spinous process;Left acromion;Right acromion;Left inferior scapulae angle;Right inferior scapulae angle;Interscapular vertebral spinous process located midway between the left and right inferior scapular angles;Vertebral spinous process located midway between points 1 and 6;Left posterior superior iliac spine;Right posterior superior iliac spine;Pelvic shelf (usually L5);Vertebral spinous process located midway between points 6 and 10.

The BHOHB device then took two back photos for each exam: the first while the patient was standing (Figure 3) and the second with the patient in the Adams test position (Figure 4).

Finally, the BHOHB software provided a 3D reconstruction of the spine (Figure 5).

The AP X-ray view was used to determine the patient’s curve magnitude (Cobb’s method), and the end vertebrae were preselected to minimize interobserver error.

### 2.4. Potential Bias

The main bias of our study could be related to the human evaluation of considered variables. To reduce this risk, every single evaluation was performed independently by 2 investigators (two orthopaedists) and was repeated after 3 months.

Another bias could lie in the flowed time between X-ray examination and BHOHB evaluation. To reduce this risk, the first evaluation was performed within two months of the X-ray.

### 2.5. Statistical Assessment

#### 2.5.1. Dimension of Sample

A previous study comparing Cobb angle measurements obtained by the Scolioscan instrument versus the gold standard showed a mean difference of 0.2° between these two methods [31]. Therefore, in this study, we assumed the same mean difference between the two methods compared (X-ray vs. BHOHB), a standard deviation of 1.85°, and a maximum allowed difference of 5°, reported as clinically relevant [32]. Accordingly, a total of 66 patients was calculated as sufficient to evaluate the correlation between the two methods with an 80% alpha error and an 80% study power.

Twenty per cent of dropped-out patients, who declined to participate in the second BHOHB examination, had to be included in our calculations. As a result, the minimum number of patients to be included in the study was 79.

#### 2.5.2. Statistic Plane

For the main end point, the validity of the BHOHB technology was estimated using the Bland–Altman method and the Pearson correlation coefficient. Study results (Cobb angle value) were expressed as average and standard deviations; values obtained with our two different methods (X-ray and BHOHB) were compared using Student’s *t*-test for paired data.

The secondary end points considered intraoperator (measurements performed by the same orthopedist but at different times) and interoperator (measurements performed by different orthopedists during the same examination) reliability. They were evaluated by the intraclass correlation coefficient (ICC).

The correlation between the BHOHB and radiography measurements was expressed by the Pearson correlation coefficient.

In both cases, the correlation coefficient value was evaluated as follows:0.25–0.50: poor correlation;0.50–0.75: moderate to good correlation;0.75–1.00: very good to excellent correlation.

Finally, ICC values were evaluated according to Currier’s criteria as follows: very reliable (0.80–1.0), moderately reliable (0.60–0.79), and questionable reliability (≤0.60).

## 3. Results

A total of 95 out of 113 initially chosen patients were enrolled in the study. A total of 18 patients were excluded from the study: 12 because they did not meet the inclusion criteria—8 showed a higher Body Mass Index, and 4 had negative Adams test results— and 6 because they did not complete the examination at t1.

The mean age among all the enrolled patients was 12.51 years. The scoliosis magnitude was 20.54 ± 9 Cobb degrees. Measurements in Cobb degrees acquired at t0 with BHOHB were 20.87° (±9.1 SD) for the first operator, while those of the second operator were 20.82° (±8.67 SD). At t1, the values were 20.23° (±8.93 SD) for the first operator and 19.90° (±8.96 SD) for the second operator.

The average prominence measured with the scoliometer was 6.15 degrees (±3.49 SD), and that with the humpmeter was 6.11 mm (±5.14 SD), while that measured with the BHOHB was 6.58 mm (±4.94 SD). The relationship between BHOHB and humpmeter was significant (r = 0.94; *p* < 0.0001). A significant correlation for prominence was also confirmed between measurements taken with the scoliometer in degrees and in mm, with r = 0.86 (*p* < 0.0001).

The relationship between the values obtained with BHOHB technology and radiography, carried out in double control and by both operators, was very high. In fact, both the Lin concordance coefficient and the Pearson correlation coefficient always assume values higher than 0.9 (*p* < 0.001), above all X-ray vs. first physician at t0, r = 0.971; X-ray vs. second physician at t0, r = 0.945; X-ray vs. first physician at t1, r = 0.886; and X-ray vs. second physician at t1, r = 0.907 (Table 1).

A very good correlation was also confirmed between clinical and machine prominence measurements: r = 0.94 (*p* < 0.0001) at t0. For the second prominence measurement taken by the first operator, slightly lower values of the Lin agreement and Pearson correlation coefficients were obtained, with r equal to 0.734 and 0.762, respectively (*p* < 0.001). Nevertheless, the results show high agreement.

Furthermore, the examination of corresponding graphs by Bland and Altman show excellent agreement between the two series of measurements. In fact, the points, representing the differences between measurements, are generally concentrated around the line of average differences and the line of perfect agreement, and they almost entirely fall within the 95% confidence band of the differences in the means (Figure 6).

Inter- and intraoperator reliability were also very consistent, evaluated through the intraclass correlation coefficient (ICC), which has always reached values higher than 0.9 (*p* < 0.001). In particular, the intraoperator ICC values were 0.986 for the first operator and 0.976 for the second. In addition, the interoperator reliability was very reliable for both the first examination at t0 (r = 0.99) and the second check at t1 (r = 0.98). The results were also confirmed by Pearson’s correlation coefficient, with values greater than 0.88 (*p* < 0.001) (Table 2, Figure 7).

## 4. Discussion

The use of surface topography for the imaging of spinal pathologies has been growing in recent years. This system is especially useful for the follow-up of patients with scoliosis since it is a radiation-free technique. In this study, we introduced a new system, and the main aim was to validate the use of the BHOHB device in the evaluation and measurement of scoliosis in comparison with the standard X-ray method. The second goal consisted of assessing intra- and interoperator reliability and evaluating the relationship between prominence measured with the BHOHB and the humpmeter.

Several studies showed that ST had high repeatability and high intraobserver and interobserver correlation, and the correlation between ST parameters and radiographic Cobb angle ranged from moderate to high [33,34].

Our results confirmed that the ICC for intraoperator and interoperator parameters were very reliable (0.80–1.0). The lack of wide variations between the first and second checks indirectly confirmed the ease of marker placement. However, the high reliability in the marker positions should not indicate that it is not important to apply them in the correct position.

In addition, our results confirmed the system’s reliability, even compared to other studies in which the Cobb ICC angle varied from moderate to high (0.61–0.78). In particular, high correlations were observed in thoracic and moderate correlations in thoracolumbar and lumbar scoliosis. Some authors explained that the reason for the moderate correlation in lumbar scoliosis might be related to the structure of the pelvic area and the placement of markers using artificial intelligence [34]. In fact, some ST systems require the operator to manually place the marker points at the specified anatomical locations to increase the accuracy of the measurement, and in many cases, it is not easy to accurately palpate or locate the marker.

Therefore, another important identified finding in comparison with the literature is that the marker-based system used in this study reduces the risk of patient posture error, because posture does not change landmarks [28,30,34]. This problem is quite present in measurements obtained by Moiré topography methods. In fact, in these systems, the posture of the body should be evaluated at the same time of day by a qualified person (physiotherapist, doctor, radiologist, or teacher) with several years of experience and always in a darkened room [34,35]. A serious problem, reported in the literature, is the occurrence of a large number of false-positive results. The high rate of false-positives was probably due to the fact that the control samples included patients referred for possible scoliosis, most of whom had some kind of asymmetry in their backs that was detected by Moiré topography methods [34].

Another important point for the screening of idiopathic scoliosis was the prominence measurement. Prominence is the most evident clinical sign of scoliosis and represents an important prognostic factor in terms of curve progression. However, many studies in the past questioned the existence of a correlation between clinical deformity and curve severity. Recent literature has shown that the hump, measured by a humpmeter, was significant for a threshold of 5 mm. Moreover, a trunk rotation angle of ≥7° for thoracic and right convex curves and ≥6° for thoracolumbar, lumbar, and left convex curves is considered a reliable criterion for identifying patients with Cobb angles of 25° or greater, thus reducing the need for spinal radiography [36].

The results of our study demonstrate how clinical prominence measured in mm correlates with measurement by BHOHB in degrees. Thus, it could be possible to use this examination for preliminary assessments without the need for an experienced operator.

The results are also consistent with studies carried out with a similar system by Goh et al. [33], where ICC values ranged from 0.98 to 0.99, whereas slightly lower results were demonstrated by Guidetti et al. [28], where ICC values ranged from 0.74 to 0.59.

Therefore, using this system to check the prominence pattern can be beneficial for performing wide-range screenings.

However, in patients with a lumbar curve, several studies showed no significant correlation between the prominence dimension and curve severity. This lack of correlation reduces the effectiveness of prominence measurement as a parameter in screenings when a lumbar curve is present [36].

In any case, the results of the study allow us to state that with surface topography, we can evaluate two parameters, the hump and the Cobb degrees, and using this method is recommended before prescribing an X-ray for the patient. This double measurement allows us to have an assessment of not only the curve but also rotation, which is very important because scoliosis is a 3D deformity. This enables us to have a complete assessment and significantly reduce the number of unnecessary radiographic examinations.

However, this study presented some limitations that could have affected the ICC values. The first is that enrolled patients were not undergoing brace treatment. The brace could alter the spine aesthetically and thus the efficiency of collected data. The second limitation is that this system operates only on the frontal plane, while scoliosis is a deformation on three space planes.

Researchers should discuss the results and how they can be interpreted from the perspective of previous studies and of the working hypotheses. The findings and their implications should be discussed in the broadest context possible, and future research directions should be highlighted.

## 5. Conclusions

This study confirmed that the results obtained by BHOHB are comparable to those obtained by X-ray. Moreover, the operator does not influence these results.

Furthermore, we can state that ST can be helpful in the diagnosis and treatment of scoliosis. In particular, it is a safe and cost-effective tool for long-term surveillance of scoliosis and early detection of progressive disease.

Although our results demonstrate an excellent correlation between radiographic examination and BHOHB examination, clinical practice for now suggests using BHOHB mainly in evaluating curve evolutions (follow-up). In fact, surface topography will not completely replace X-ray analysis in scoliosis patients because it cannot assess real bone morphology in the same way as an X-ray.

However, there are obvious advantages to not repeating X-rays in teenagers, such as reducing exposure to ionizing radiation.

If surface topography can provide reliable and comparable results, it should replace X-rays at clinical visits when curve monitoring is required to prevent unnecessary radiation exposure.

Identifying topographic changes allows us to select patients who should be referred for radiographic imaging to confirm curve progression and determine therapeutic intervention.

## Figures and Tables

**Figure 1 children-10-00320-f001:**
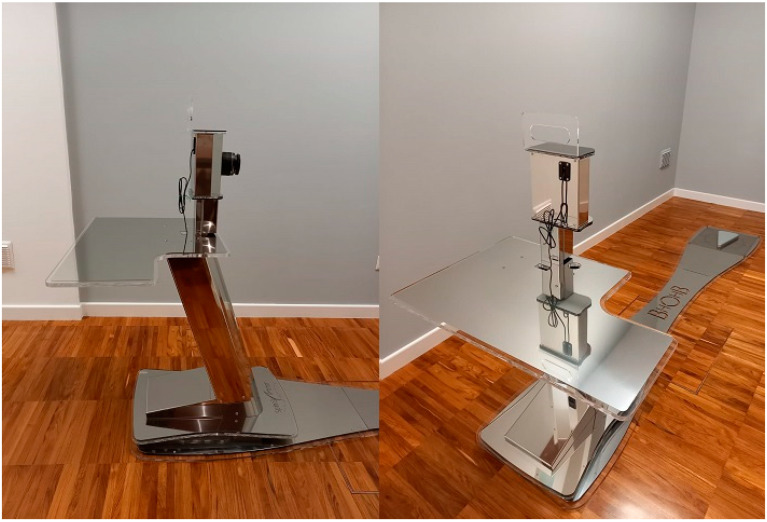
Biometrical Holistic Of Human Body (BHOHB) machine.

**Figure 2 children-10-00320-f002:**
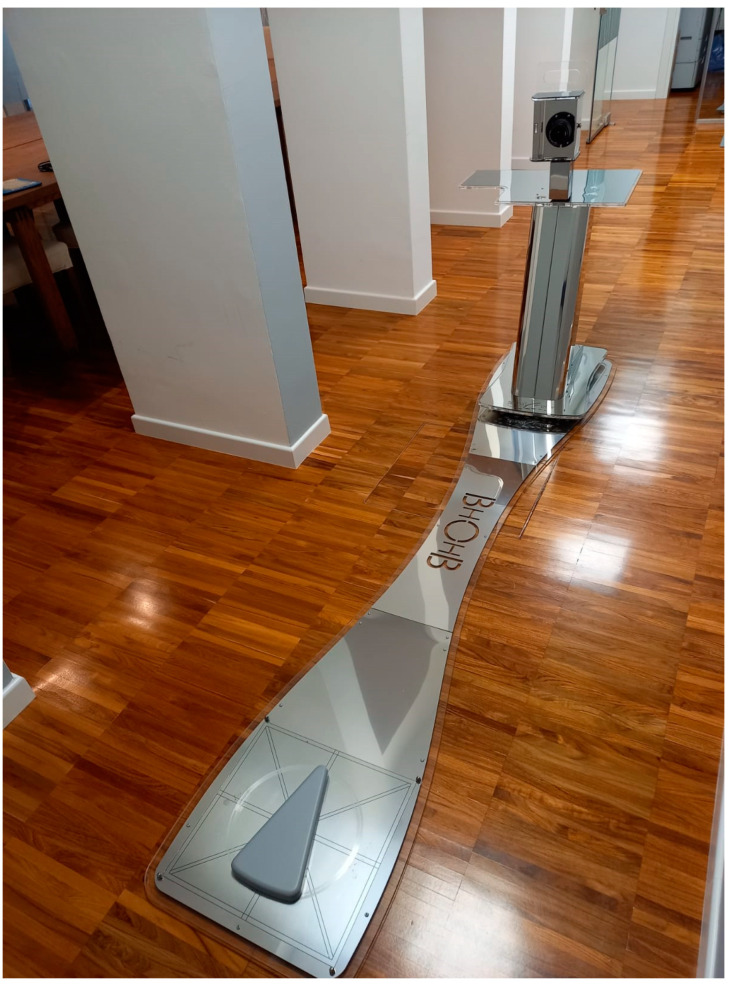
BHOHB platform.

**Figure 3 children-10-00320-f003:**
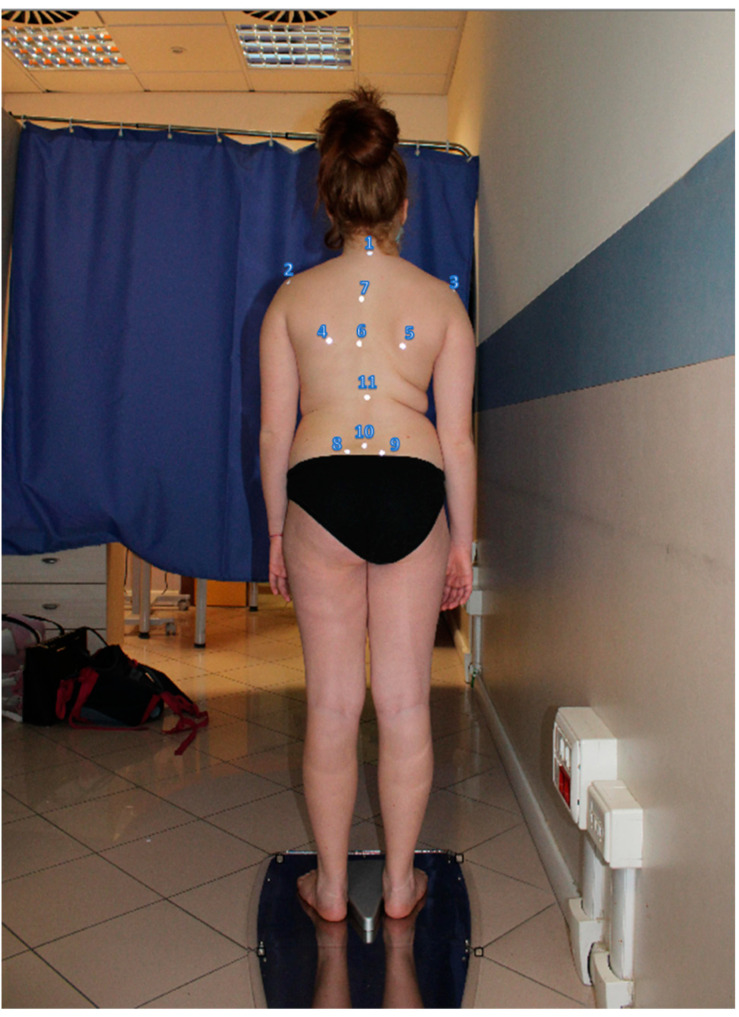
Set point positions.

**Figure 4 children-10-00320-f004:**
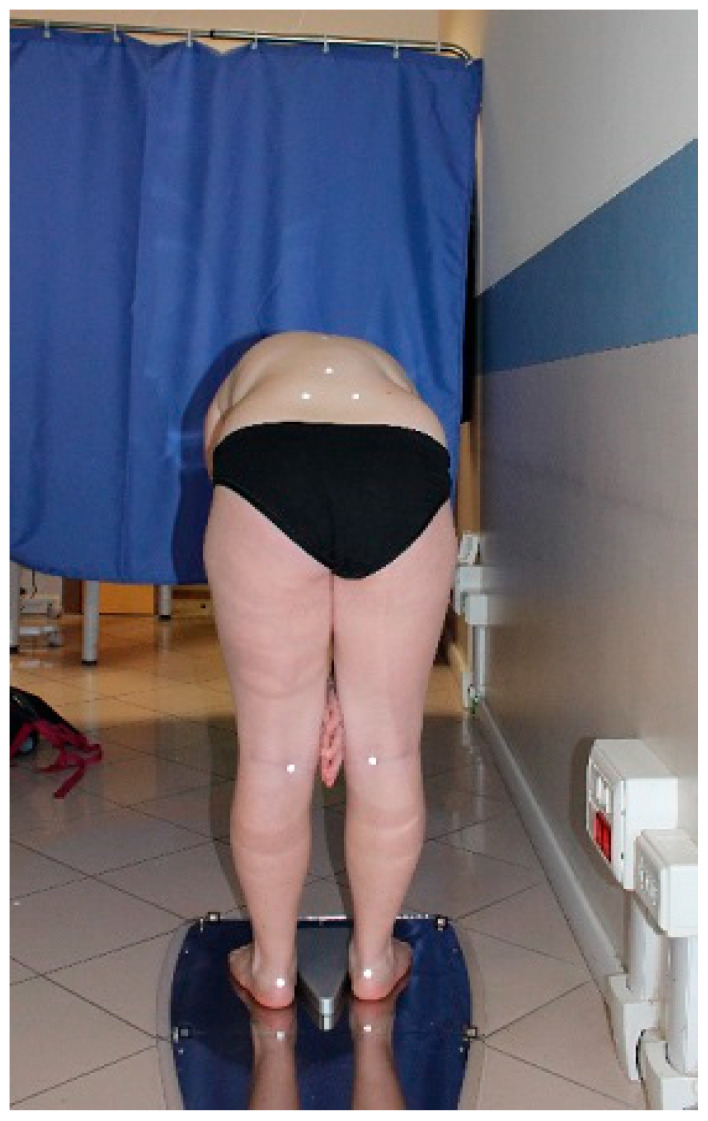
The prominence in an Adams test.

**Figure 5 children-10-00320-f005:**
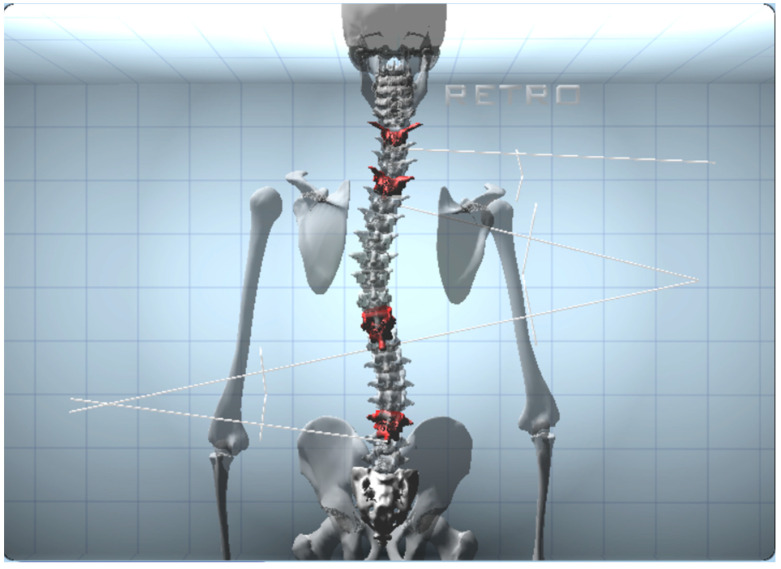
Three-dimensional BHOHB spine reconstruction.

**Figure 6 children-10-00320-f006:**
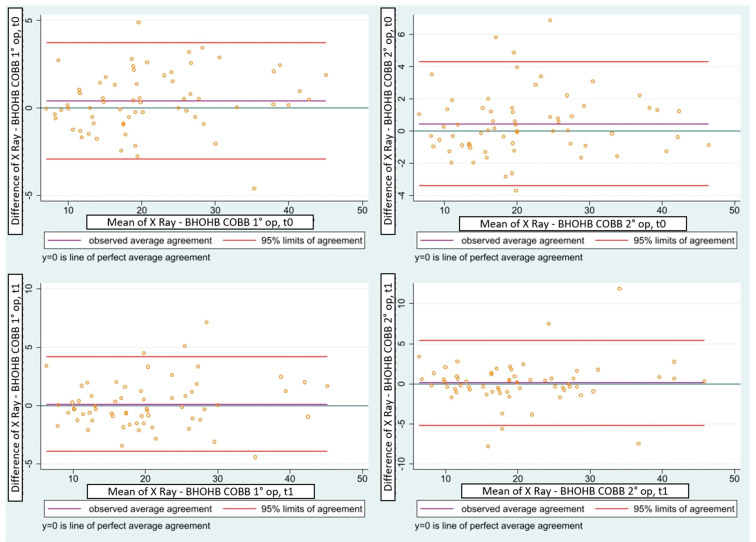
Concordance correlation confidence (Lin 1989–2000) between Cobb and BHOHB for the first and second operators in t0 and t1.

**Figure 7 children-10-00320-f007:**
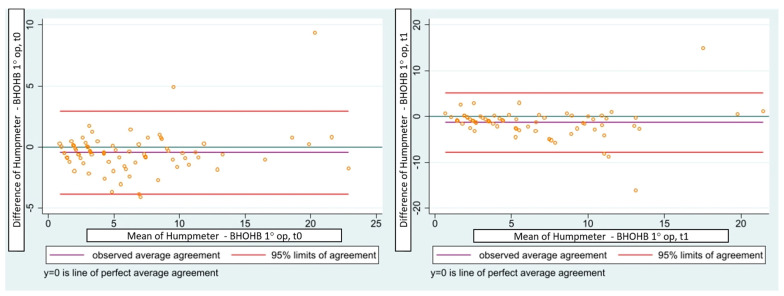
Concordance correlation confidence (Lin 1989–2000) between the humpmeter and BHOHB in t0 and t1.

**Table 1 children-10-00320-t001:** Correlation between X-ray, humpmeter, and BHOHB.

	LIN Concordance Correlation Coefficient (*p*-Value)	Pearson Correlation Coefficient (*p*-Value)
X-ray-BHOHB COBB 1° op, t0	0.971 (<0.001)	0.972 (<0.001)
X-ray-BHOHB COBB 2° op, t0	0.945 (<0.001)	0.955 (<0.001)
X-ray-BHOHB COBB2 1° op, t1	0.886 (<0.001)	0.903 (<0.001)
X-ray-BHOHB COBB2 2° op, t1	0.907 (<0.001)	0.927 (<0.001)
Humpmeter-BHOHB 1° op, t0	0.939 (<0.001)	0.945 (<0.001)
Humpmeter-BHOHB 1° op, t1	0.734 (<0.001)	0.762 (<0.001)

0.25–0.50: poor correlation, 0.50–0.75: moderate to good correlation, 0.75–1.00: very good to excellent correlation.

**Table 2 children-10-00320-t002:** Inter- and intraoperator reliability evaluation with ICC and Pearson.

Reliability Intraoperator	Intraclass Correlation Coefficient (ICC)	Pearson Correlation Coefficient (*p*-Value)
t0–1st op	0.986 (<0.001)	0.973 (<0.001)
t1–1st op	0.935 (<0.001)	0.884 (<0.001)
t0–2nd op	0.976 (<0.001)	0.954 (<0.001)
t1–2nd op	0.970 (<0.001)	0.943 (<0.001)
**Reliability interoperator**		
t0: 1st op–2nd op	0.990 (<0.001)	0.980 (<0.001)
t1: 1st op–2nd op	0.980 (<0.001)	0.962 (<0.001)

## Data Availability

Datasets generated and/or analyzed during the current study are available from the corresponding author upon reasonable request.

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
