# Peer review of "Is Surface Topography Useful in the Diagnosis of Scoliosis? Validation of the Biometrical Holistic of Human Body (BHOHB)"

_children, 2023, doi:10.3390/children10020320_

Round 1

Reviewer 1 Report

Dear Authors,

 I would like to congratulate yourself for the developed research. The theme of your study is highly relevant. Please consider the appointments below.

 ·         The key words “Surface topography” and “BHOHB” are not descriptors in MESH (National Library of Medicine). I suggest changing them in order to facilitate futures bibliographic database searching.

·       ·    The World Health Organization (WHO) defines 'Adolescents' as individuals in the 10-19 years age group. Therefore, I suggest the use of the term “adolescent” instead of “children” considering the age group studied in this research.

·     ·      How did you assess the positivity of the Adams test? Qualitatively? By the scoliometer or the humpmeter? What was the cut off consider for the positivity?

·       ·    Why did you exclude the individuals with BMI higher than 25kg/m2? Are there some limitations about the use of BHOHB in this population?

·       ·    The acronym BMI needs to be explained the first time it appears in the text.

·       ·     Considering the anthropometric variability of the individuals, why did you prefer to choose the middle of the way to locate some of vertebral spinous process instead a specific vertebral spinous process? Do you think this methodology could influence the comparison between the BHOHB and X-rays measurements?

·      ·     Did you use the same methodology to measure the Cobb angle in both BHOHB and the X-rays? For example, did you consider the same apical, upper end and lower end vertebrae in both BHOHB and X-rays for the same volunteer? Please, give more information about measuring the Cobb angle in the studied instruments.

·     ·      In table 1, there is a variable humpmeter – BHOHB. Why did only one operator assess this variable?

·      ·     It is necessary to replace the comma with a point in the decimal numbers in table 1.

·      ·     In this sentence: “In particular: X-rays vs 1st operator at t0, r = 0.983; X-193 rays vs 2nd operator at t0, r = 0.978; X-rays vs 1st operator at t1, r = 0.971; X-rays vs 2nd  operator at t1, r = 0.945”, X-rays vs what? I didn´t find the correspondence of these results in table 1…Please, I need more detail to understand this sentence.

·      ·     How did you do the prominence measurement? Please describe how you used the humpmeter to measure this variable. For example, did you consider only one prominence measurement? What angle of the forward bending test was considered to the prominence measurement?

·        ·   Please check this information: “Only for the second check prominence measurement took by the second operator…”. Table 1 contains only the description of the first operator.

·      ·     If possible, in figure 6, standardize operator and evaluation time abbreviations with the ones presented in table 1 and in the text. The same standardization is necessary for figure 7.

·        ·   Please give more information about this sentence: “Another important identified finding in comparison with the literature is that the marker-based system used in this study reduces the risk of patient posture error”.

·        ·   References: Only 17.6% of references are from the last 10 years. If possible, it would be important to update them.

 Kind regards.

Author Response

Dear editor and reviewers thanks for examining our paper and found it interesting.

We have read carefully the comments of reviewers and we followed their instructions to make some changes to the text. 

  • The key words “Surface topography” and “BHOHB” are not descriptors in MESH (National Library of Medicine). I suggest changing them in order to facilitate futures bibliographic database searching.

We have changed in the key words, surface topography in topography.

  • The World Health Organization (WHO)defines 'Adolescents' as individuals in the 10-19 years age group. Therefore, I suggest the use of the term “adolescent” instead of “children” considering the age group studied in this research.

I agree with this suggestion and we have changed children to adolescent when we were talking about a population over 10 years old.  “The primary aim of this study was to validate the new BHOHB technology in adolescent with scoliosis among the gold standard X-ray method.”

  • How did you assess the positivity of the Adams test? Qualitatively? By the scoliometer or the humpmeter? What was the cut off consider for the positivity?

We add in the text “The Prominence was measured with the Scoliometer (degrees) and with the Hump-meter (millimeters) in the Adams Test position”

  • Why did you exclude the individuals with BMI higher than 25kg/m2? Are there some limitations about the use of BHOHB in this population?

Since the markers are applied clinically, we did not want bias due to the difficulty of finding the anatomical landmarks. The next step will be studying them.

  • The acronym BMI needs to be explained the first time it appears in the text.

We have changed “Patients with a BMI (Body Mass Index) higher than 25.0 kg/m2”

  • Considering the anthropometric variability of the individuals, why did you prefer to choose the middle of the way to locate some of vertebral spinous process instead a specific vertebral spinous process? Do you think this methodology could influence the comparison between the BHOHB and X-rays measurements?

The position of the markers has been studied by the engineer who developed the software to reduce the the measurement error.

  • Did you use the same methodology to measure the Cobb angle in both BHOHB and the X-rays? For example, did you consider the same apical, upper end and lower end vertebrae in both BHOHB and X-rays for the same volunteer? Please, give more information about measuring the Cobb angle in the studied instruments.

We add in the text “The AP X-Ray view was used to determine the patient’s curve magnitude (Cobb’s method) and the end-vertebrae were pre-selected to minimize the interobserver error”.

  • In table 1, there is a variable humpmeter – BHOHB. Why did only one operator assess this variable?

The measurement of the prominence made with the bhohb is of a single operator because it is independent of the markers and therefore does not change between the operators.

  • It is necessary to replace the comma with a point in the decimal numbers in table 1.

We have changed it

  • In this sentence: “In particular: X-rays vs 1st operator at t0, r = 0.983; X-193 rays vs 2nd operator at t0, r = 0.978; X-rays vs 1st operator at t1, r = 0.971; X-rays vs 2nd  operator at t1, r = 0.945”, X-rays vs what? I didn´t find the correspondence of these results in table 1…Please, I need more detail to understand this sentence.

We have reported in the text wrong data, coming from an incomplete analysis performed at the start of the study. We have changed the text. “In particular: X-ray vs 1st operator at t0, r = 0.971; X-ray vs 2nd operator at t0, r = 0.945; X-ray vs 1st operator at t1, r = 0.886; X-ray vs 2nd operator at t1, r = 0.907”

  • How did you do the prominence measurement? Please describe how you used the humpmeter to measure this variable. For example, did you consider only one prominence measurement? What angle of the forward bending test was considered to the prominence measurement?

We add in the text the sentence: “In this test the patient bend forward, starting at the waist until the observed prominence was parallel with the floor, with the feet together, arms hanging and the knees in extension. The palms are held together and placed between the knees. The operator moved the scoliometer up and down the prominence until the largest ATR measurement was achieved and likewise was evaluated with the humpmeter.”

  • Please check this information: “Only for the second check prominence measurement took by the second operator…”. Table 1 contains only the description of the first operator.

We change the text: “Only for the second check prominence measurement took by the first operator…”.

  • If possible, in figure 6, standardize operator and evaluation time abbreviations with the ones presented in table 1 and in the text. The same standardization is necessary for figure 7.

We have changed it.

  • Please give more information about this sentence: “Another important identified finding in comparison with the literature is that the marker-based system used in this study reduces the risk of patient posture error”.

We add in the text: “because posture doesn't change landmarks”

  • References: Only 17.6% of references are from the last 10 years. If possible, it would be important to update them.

We add another reference.

Reviewer 2 Report

The presented piece of research meets the criteria of the Children's Journal. BHOHB could be an interesting method with regard to the conservative treatment of scoliosis and the reduction of x-ray exposure. 

Author Response

Dear reviewer thanks for examining our paper and found it interesting. 

Reviewer 3 Report

Dear Author, 

Article Is Surface topography useful in the diagnosis of scoliosis? Vali- 2 dation of the Biometrical Holistic Of Human Body (BHOHB) is very intresting. 

Author Response

(The authors gave the same response as above.)
